# Tunable inverted gap in monolayer quasi-metallic MoS$_2$ induced by strong charge-lattice coupling

Xinmao Yin [1,2,3], Qixing Wang[2], Liang Cao [2,4], Chi Sin Tang[2,5], Xin Luo[2,6,7], Yujie Zheng[2], Lai Mun Wong[8], Shi Jie Wang[8], Su Ying Quek[2,6], Wenjing Zhang[1], Andrivo Rusydi[2,3,5,9] & Andrew T.S. Wee [2,5,6]

Polymorphism of two-dimensional transition metal dichalcogenides such as molybdenum disulfide (MoS$_2$) exhibit fascinating optical and transport properties. Here, we observe a tunable inverted gap (~0.50 eV) and a fundamental gap (~0.10 eV) in quasimetallic monolayer MoS$_2$. Using spectral-weight transfer analysis, we find that the inverted gap is attributed to the strong charge–lattice coupling in two-dimensional transition metal dichalcogenides (2D-TMDs). A comprehensive experimental study, supported by theoretical calculations, is conducted to understand the transition of monolayer MoS$_2$ on gold film from trigonal semiconducting 1H phase to the distorted octahedral quasimetallic 1T' phase. We clarify that electron doping from gold, facilitated by interfacial tensile strain, is the key mechanism leading to its 1H–1T' phase transition, thus resulting in the formation of the inverted gap. Our result shows the importance of charge–lattice coupling to the intrinsic properties of the inverted gap and polymorphism of MoS$_2$, thereby unlocking new possibilities for 2D-TMD-based device fabrication.

[1] SZU-NUS Collaborative Innovation Center for Optoelectronic Science & Technology, Key Laboratory of Optoelectronic Devices and Systems of Ministry of Education and Guangdong Province, College of Optoelectronic Engineering, Shenzhen University, Shenzhen 518060, China. [2] Department of Physics, Faculty of Science, National University of Singapore, 117542 Singapore, Singapore. [3] Singapore Synchrotron Light Source (SSLS), National University of Singapore, 117603 Singapore, Singapore. [4] Anhui Province Key Laboratory of Condensed Matter Physics at Extreme Conditions, High Magnetic Field Laboratory of the Chinese Academy of Sciences, Hefei 230031, China. [5] NUS Graduate School for Integrative Sciences and Engineering, National University of Singapore, 117456 Singapore, Singapore. [6] Centre for Advanced 2D Materials and Graphene Research Centre, National University of Singapore, 117551 Singapore, Singapore. [7] Department of Applied Physics, The Hong Kong Polytechnic University, Hung Hom, Kowloon, Hong Kong 999077, China. [8] Institute of Materials Research and Engineering (IMRE), A*STAR (Agency for Science, Technology and Research), 2 Fusionopolis Way, Innovis 138634, Singapore. [9] NUSNNI-NanoCore, National University of Singapore, Singapore 117576, Singapore. Qixing Wang and Liang Cao contributed equally to this work. Correspondence and requests for materials should be addressed to W.Z. (email: wjzhang@szu.edu.cn) or to A.R. (email: phyandri@nus.edu.sg) or to A.T.S.W. (email: phyweets@nus.edu.sg)

Exotic electronic phases arising from the intricate coupling between charge, spin, orbital and lattice degrees of freedom have brought about much attention and interest in condensed matter physics[1–3]. Two-dimensional (2D) molybdenum disulfide ($MoS_2$), a semiconductor with direct bandgap[4], high mobility and room-temperature on/off current ratio[5], has attracted much attention due to its potential applications in spintronics, valleytronics, optoelectronics and nanoelectronics[5–7].

Similar to graphene, monolayer $MoS_2$ can be extracted from bulk crystals by chemical, mechanical exfoliation[8] or grown into large-area samples via chemical vapor deposition (CVD)[9]. While the trigonal semiconducting 1H phase of $MoS_2$ is desired for the aforementioned applications, the octahedral metallic 1T phase has applications in areas such as supercapacitor electrodes[10], hydrogen evolution reaction catalysts[11, 12] and Weyl semimetals[13]. In this regard, controlling the phase composition of

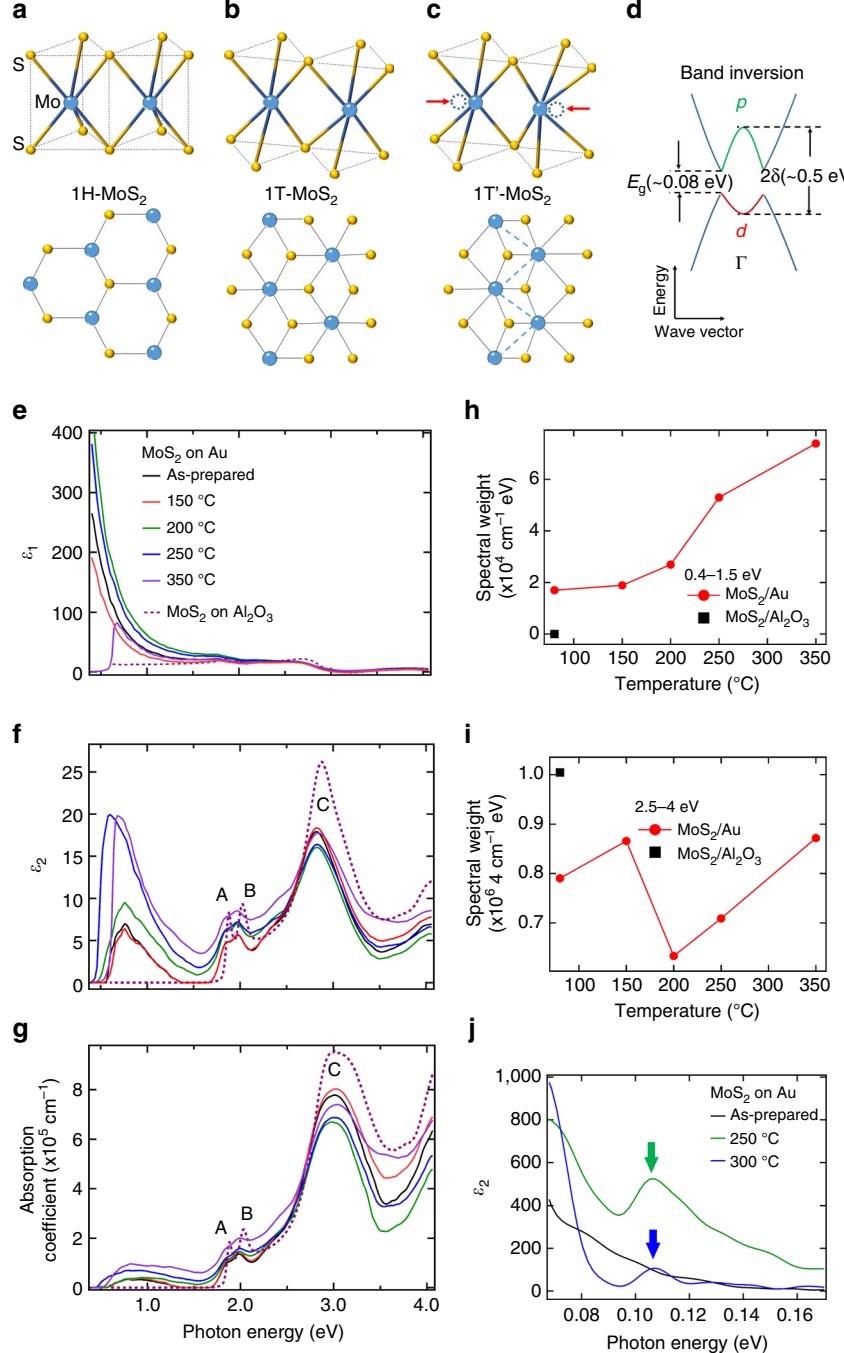

**Fig. 1** Atomistic and band structures of monolayer $MoS_2$ and annealing temperature dependence of optical spectra. **a–c** Structure of 1H-$MoS_2$, 1T-$MoS_2$ and 1T'-$MoS_2$. **d** Schematic band structure of 1T'-$MoS_2$. $E_g$-fundamental gap; $2\delta$-inverted gap. **e, f** Dielectric functions $\varepsilon_1$ and $\varepsilon_2$ of monolayer $MoS_2$ on Au annealed at various temperatures and on $Al_2O_3$ substrate from VASE spectroscopic ellipsometry. **g** Absorption coefficient spectra. The color-coded *vertical arrows* show the shift of the threshold energy of the pre-peak from pristine (0.58 eV, *black*), 250 °C (0.46 eV, *blue*) to 350 °C (0.54 eV, *purple*). **h, i** Integrated spectral-weight $\int_{\omega_1}^{\omega_2} \alpha_1(\omega, T)\, d\omega$ in regions 0.4–1.5 and 2.5–4 eV. **j** The imaginary component of dielectric function, $\varepsilon_2$, of monolayer $MoS_2$/Au in the far-infrared regime from IR-VASE spectroscopic ellipsometry at various annealing temperatures with color-coded *arrows* indicating the position of the fundamental gap

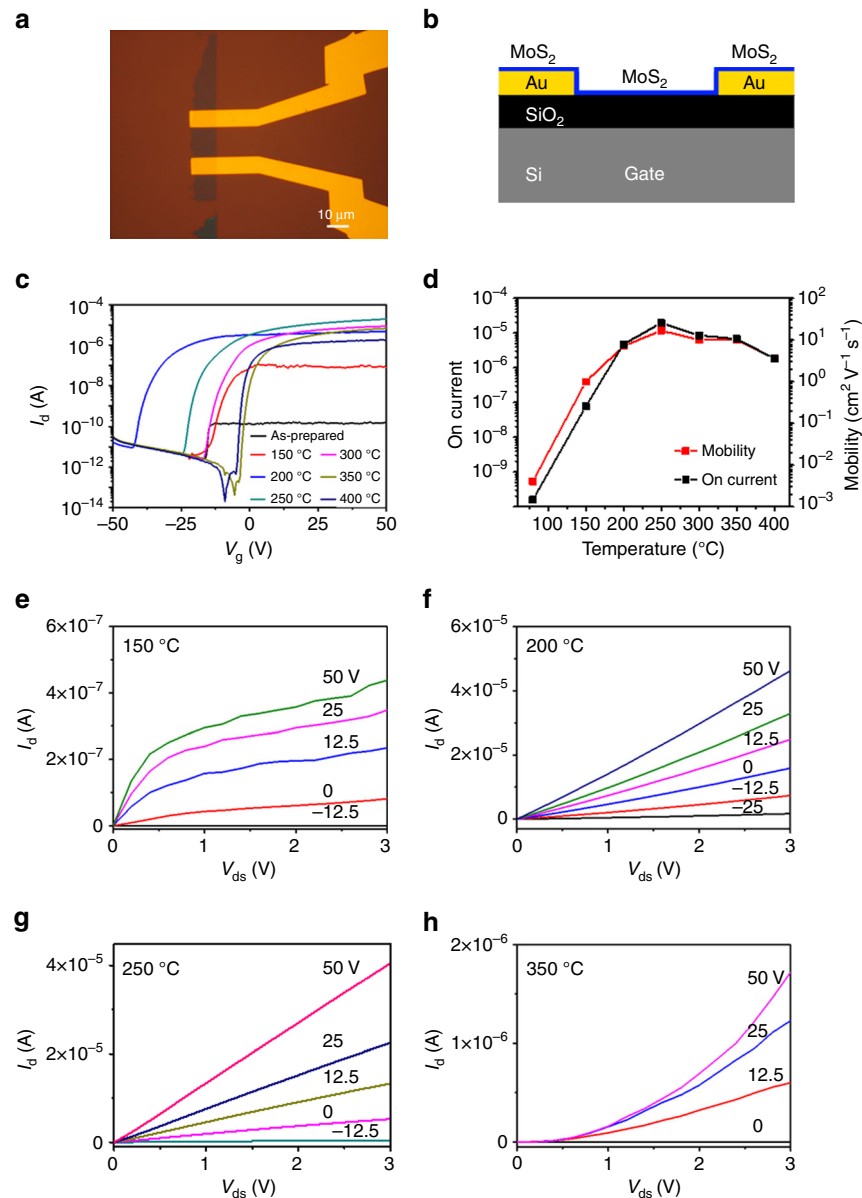

**Fig. 2** Device fabrication and annealing temperature dependence of transport properties. **a** Optical microscopy of a $MoS_2$ field-effect transistor. **b** Device structure schematic. **c** $I_d$–$V_g$ of the $MoS_2$ device as functions of annealing temperature. **d** Saturated ON current and mobility vs. temperature at different gate voltages of the device as functions of annealing temperature. **e**–**h** $I_d$–$V_{ds}$ curves of the device at respective temperatures

monolayer $MoS_2$ is important for device integration and scalable processing. The semiconductor-to-metal phase transition of 2D-transition metal dichalcogenides (TMDs) was previously demonstrated via *n*-butyl lithium (*n*-BuLi) treatment[14] by electron transfer from *n*-BuLi to TMD and the interlayer atomic plane gliding during intercalation[15–18]. Reports have shown theoretically and experimentally that the 1T structure was unstable in free-standing conditions and spontaneously relaxed to a distorted 1T' structure[19–21]. In the 1T' structure, the distorted transition metal atoms formed a period doubling 2 × 1 structure consisting of 1D zigzag chains. Raman and photoluminescence (PL) spectroscopies have been commonly used to characterize the 1T/T' phase[22, 23]. This polymorphic nature of $MoS_2$ is closely related to the coupling between charge, spin, orbital and lattice degrees of freedom.

Recently, a theoretical study predicted that the spin–orbit coupling might open a fundamental gap[20], $E_g$, in the distorted octahedral structure (1T') of 2D-TMDs, which was observed to be ~0.06 eV for few-layer 1T'-$MoTe_2$ using high-resolution Fourier-transform infrared spectroscopy[21]. Intriguingly, both density functional theory (DFT) and GW first-principles calculations predicted a larger mid-infrared inverted gap $2\delta$ (~0.6 eV) at the $\Gamma$-point in the 2D Brillouin zone of the 1T' structure. However, this inverted gap is yet to be experimentally observed. Furthermore, the origin of the inverted gap is not clear. On one hand, DFT calculations suggested that the inverted gap and its low energy electronic structure might originate from the distorted octahedral structure. On the contrary, GW-based calculations suggested the importance of many-body effects of electron–electron interactions. Therefore, it is important to measure and clarify the origin of the inverted gap and the low energy electronic structure because they directly affect our fundamental understanding of 2D-TMDs in general.

In this work, we observe a mid-infrared peak at ~0.5 eV (inverted gap) and a far-infrared peak at ~0.10 eV (fundamental gap) in annealed monolayer $MoS_2$ on gold film ($MoS_2$/Au) using

spectroscopic ellipsometry. Based on a comprehensive study involving transport, Raman, PL and synchrotron-based photo-emission spectroscopy (PES), supported by theoretical calculations, we monitor the 1H-to-1T' phase transition in monolayer $MoS_2$/Au upon annealing and study the changes in its optical and electronic properties. Detailed analysis shows that the 1H–1T' phase transition is due to electron transfer from gold to monolayer $MoS_2$. While this underlying mechanism is somewhat similar to the previously reported $n$-BuLi treatment method[14], the phase transition of monolayer $MoS_2$ on Au is further facilitated by interfacial strain. This is a convenient and straightforward way of inducing a semiconductor-to-metal phase transition in 2D-$MoS_2$ based on an annealing process. This technique has potential applications such as the fabrication of 2D-TMD-based field-effect transistors. Our analysis shows that the mid-infrared peak corresponds to the 1T'-$MoS_2$ inverted gap which is tunable by interfacial strain, while the far-infrared peak corresponds to the fundamental gap. Based on spectral-weight transfer analysis, we find that this inverted gap can be attributed to a combination of the distorted lattice and electron–electron correlations in 2D-TMDs. This demonstrates the presence of a strong charge–lattice coupling in 1T' 2D-TMDs and is different from the fundamental gap which is attributed to the presence of strong spin–orbit coupling. In comparison with spin–orbit coupling, the investigation of charge–lattice coupling and its effect on the inverted band structure in 2D-TMDs may also provide a new understanding in the study of topological insulating phases.

## Results

**Optical properties of monolayer $MoS_2$.** Figure 1 displays the molecular and electronic structures of monolayer $MoS_2$ in its specific phases and the annealing temperature dependence of the optical spectra. In particular, Fig. 1e, f show the dielectric functions ($\varepsilon(\omega) = \varepsilon_1(\omega) + i\varepsilon_2(\omega)$) of $MoS_2$ on Au film on $SiO_2$/Si substrate ($MoS_2$/Au) and on sapphire ($MoS_2$/$Al_2O_3$) respectively, with the absorption coefficient spectra in Fig. 1g, measured by spectroscopic ellipsometry. Above 1.5 eV, $\varepsilon_2$ and the absorption coefficient of $MoS_2$/$Al_2O_3$ consists of three main peaks—A, B and C. They are related to the excitonic transitions[4] at the K/K' points (A and B) and the strong interband optical transition (broad C feature) due to band nesting[24] between K and $\Gamma$ in the Brillouin zone of 1H-$MoS_2$.

When $MoS_2$ is transferred onto Au, peaks A and B are red-shifted and their intensities reduced. This is evidence for electron transfer from gold to $MoS_2$[25]. Upon annealing, the increase in low-energy spectral-weight (<1.5 eV) of the mono-layer $MoS_2$/Au (Fig. 1h) indicates a rise in effective electron number[26, 27] in $MoS_2$. This suggests an increase of electron transfer from gold (>200 °C). This is further supported by a theoretical study which indicates the effect of electron doping of $MoS_2$ on Au surface[28]. Peak C shows a reduced intensity when $MoS_2$ is transferred onto Au from $Al_2O_3$. This peak is further reduced at ~200 °C (Fig. 1f, g, i) due to the effects of phase transition to be discussed later.

Interestingly, $\varepsilon_2$ and the absorption spectra show a broad mid-infrared pre-peak at the low-energy region (~0.6–1.5 eV) of $MoS_2$/Au. This is absent from $MoS_2$/$Al_2O_3$. As monolayer $MoS_2$ is transferred from $Al_2O_3$ substrate to Au film, the spectral weight at the high-energy region (above 2.5 eV) decreases. The mid-infrared peak and the increase of spectral weight below 1.5 eV (Fig. 1h) accompanied by the decrease of spectral weight above 2.5 eV (Fig. 1i) together show a signature of strong electronic correlation[29, 30] in $MoS_2$/Au. This larger than 2eV-range spectral-weight transfer (from high to low energy region) reveals that electron–electron correlation plays an

important role in the origin of the mid-infrared gap[27, 29, 30]. This is consistent with the observed dramatic rise of $\varepsilon_1$ below 1.5 eV (Fig. 1e), which suggests an unscreened electron–electron interaction enhancing electronic correlations. The increase of $\varepsilon_1$ below 1.5 eV in $MoS_2$/Au also rules out the possibility of Au clusters in $MoS_2$ because $\varepsilon_1$ of Au is typically negative below this energy range[31]. The intensity of this pre-peak in $MoS_2$/Au increases upon annealing up to 250 °C, while the threshold energy reduces from 0.58 to 0.46 eV before returning to 0.54 eV at 350 °C (color-coded *arrows* in Fig. 1g). The observation of this new pre-peak indicates a new excitonic transition which is much lower than the direct bandgap[4]. Furthermore, another interesting peak is observed in the far-infrared regime at ~0.10 eV upon sample annealing at 250 °C as shown in Fig. 1j. This is ascribed to the fundamental gap as discussed later.

**Electronic performance of $MoS_2$-based field-effect transistor.** With this inspiration, mechanical exfoliated monolayer $MoS_2$ was transferred on prepared Au electrodes (Fig. 2a, b) to study the annealing temperature-dependence of the transport properties (Fig. 2c–h) of the $MoS_2$ field-effect transistor. As shown in Fig. 2c, the transport properties of the device are very sensitive to the annealing temperature. It shows a field-effect mobility of ~$4.1 \times 10^{-3}$ cm$^2$ V$^{-1}$ s$^{-1}$ of the as-prepared device. Upon anneal-ing at 150, 200 and 250 °C, the mobility rises significantly to 1.0, 7.2 and 17.0 cm$^2$ V$^{-1}$ s$^{-1}$ respectively. Mobility at 250 °C is higher than bottom-gated $MoS_2$ transistor on $SiO_2$/Si substrate reported in a previous transport study[5]. Thereafter, mobility dropped to 10.1, 7.1 and 3.6 cm$^2$ V$^{-1}$ s$^{-1}$ upon annealing at 300, 350 and 400 °C respectively. Correspondingly, the device saturation ON current (Fig. 2d) rises considerably from 0.16 to 78 nA, 4.8 µA and 20 µA upon annealing at 150, 200 and 250 °C, respectively. Further annealing at 300, 350 and 400 °C causes the saturation current to drop to 8.6, 5.6 and 3.6 µA, respectively. The rising trend of mobility and ON current shown in Fig. 2c, d indicates that the device performance is greatly optimized upon annealing between 200 and 250 °C. Comparing the device drain current at 200 °C (Fig. 2f), the drain current at 150 °C (Fig. 2e) is two orders of magnitude lower and is more prone to saturation. This suggests that the contact at 150 °C is Schottky type and possesses a high resistance. The linear $I_d$−$V_{ds}$ characteristic at 200–250 °C (Fig. 2f, g) indicates an ohmic contact. The $I_d$−$V_{ds}$ grows increasingly nonlinear at higher temperatures (Fig. 2h). This suggests that the contact reverts to Schottky type with increased resistance as $MoS_2$ dissociates. The high-mobility, large saturation ON current and linear $I_d$−$V_{ds}$ features show a reduced device contact resistance upon annealing up to 250 °C.

**Charge–lattice coupling-induced inverted and fundamental gaps.** As demonstrated later by PES, PL and Raman measurements, the transition from 1H to 1T' phase is optimized after annealing monolayer $MoS_2$/Au between 200 and 250 °C. Furthermore, the threshold energy of the pre-peak in Fig. 1f is close to the inverted gap (~0.5 eV) of 1T'-$MoS_2$ as calculated in ref. 20. Therefore, we argue that the excitonic transition corresponding to the inverted gap of 1T'-$MoS_2$ contributes to the pre-peak. The presence of this inverted gap is further substantiated by valence band spectra (Supplementary Fig. 1b, c and Supplementary Note 1) using PES. Supported by Raman measurements and first-principles calculations (as discussed below), the inverted gap is tunable by interfacial strain. This ability to tune the inverted gap by strain is consistent with theoretical calculations in a previous study[20]. Therefore, our work provides strong evidence that the mid-infrared peak corresponds to the inverted gap.

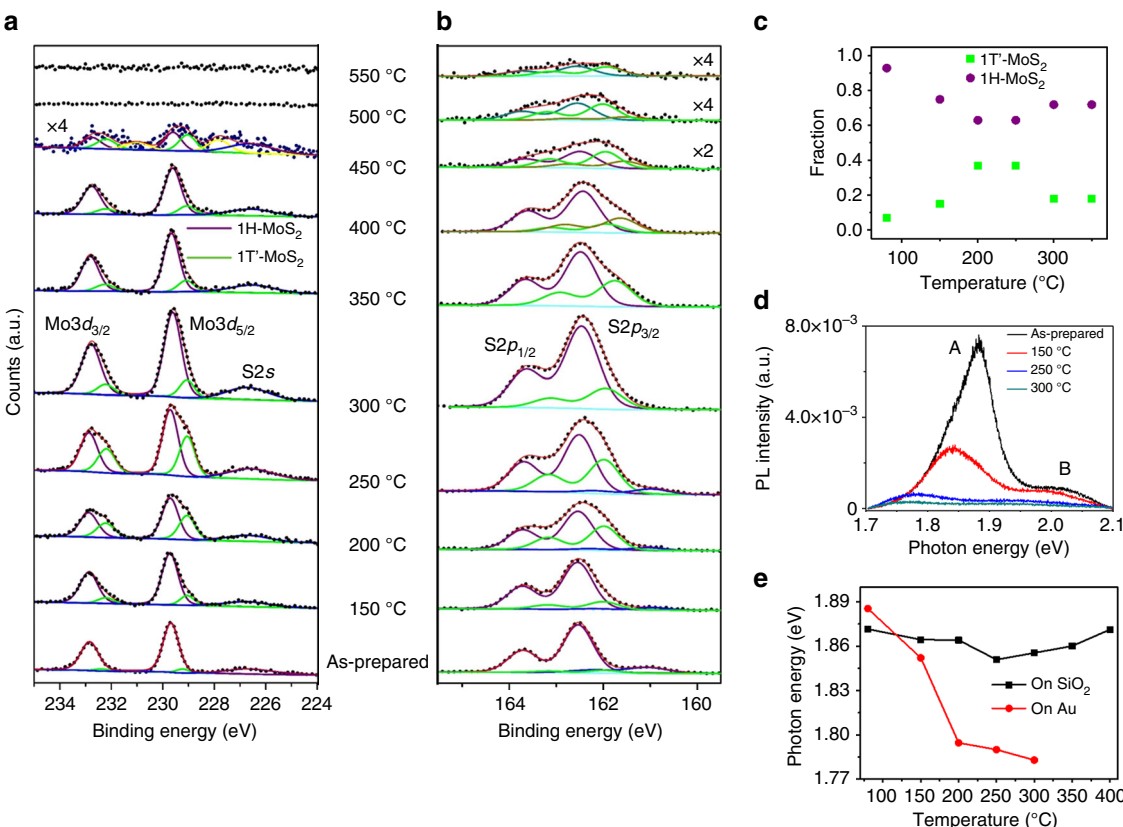

**Fig. 3** Annealing temperature dependence of photoemission and photoluminescence spectroscopy. **a** Mo-3$d$ and **b** S-2$p$ spectra for MoS$_2$/Au after annealing at respective temperatures. **c** Extracted relative fraction of 1H-MoS$_2$ and 1T'-MoS$_2$ components of Mo-3$d_{5/2}$ peaks as functions of temperature. **d** The photoluminescence spectra of monolayer MoS$_2$ on Au electrode in the device (Fig. 2b) at respective temperatures. **e** Extracted photoluminescence energy position of peak A as a function of temperature, compared with that of MoS$_2$ on SiO$_2$/Si

Moreover, the feature which appears upon sample annealing at ~0.10 eV (Fig. 1i) is associated with the fundamental gap of 1T'-MoS$_2$ (c.f., Fig. 1d). The fundamental gap is nearly annealing temperature independent after it appears. This is consistent with the previous theoretical study of monolayer MoS$_2$[20]. Experimental data in Fig. 1i also show that the Drude response enhances upon sample annealing. It is responsible for the increased mobility due to charge transfer, which will be further demonstrated.

We examine other possible explanations for the mid-infrared optical peak at ~0.5 eV in Fig. 1f. Namely, vacancies introduced during sample growth, and physisorbed impurities introduced during sample transfer. Firstly, the presence of intrinsic vacancies in CVD-grown monolayer MoS$_2$ has been reported[32, 33]. However, this can be ruled out as the cause of the ~0.5 eV optical peak because this peak only appears after the transfer of MoS$_2$ onto Au and is absent in the case of MoS$_2$/Al$_2$O$_3$[34]. Secondly, there is an increase in the ~0.5 eV optical peak intensity with increasing annealing temperature. However, sample annealing in ultrahigh vacuum results in the desorption of absorbed impurities. Hence, we can also rule out physisorbed impurities being introduced during the transfer process as a contributing factor to the ~0.5 eV mid-infrared peak.

An important observation from our spectroscopic ellipsometry data (Fig. 1f) is the significant decrease in spectral weight at the high-energy region (above 2.5 eV) when MoS$_2$ is transferred from Al$_2$O$_3$ (*dashed violet line*) to Au (*black solid line*). Even before the high-temperature annealing process, this significant drop in spectral weight cannot be explained solely by the change in 1T' lattice structure because of the low yield in 1H–1T' MoS$_2$

transition (~7% as discussed later). According to the Zaanen–Sawatzky–Allen Theory[30], spectral-weight transfers can occur over a broad energy range due to strong local charge interactions in correlated electronic systems. Therefore, the large spectral-weight transfer from the high-energy region (above 2.5 eV) to the inverted gap energy region (below 1.5 eV) reveals that electron–electron correlation must play an important role in the origin of this inverted gap. This is also supported by a previous GW-based study on 1T' 2D-TMDs[20]. Combining the predictions of this theoretical study[20] with our optical data implies the presence of charge–lattice coupling, an interplay between electron–electron correlation and lattice distortion in 1T'-MoS$_2$, which is the primary cause of this inverted gap.

The presence of the pre-peak along with peaks A, B and C is due to the coexistence of MoS$_2$in both 1H and 1T' phases which will be demonstrated by our PES data. With its immense potential for applications in device fabrication, the improvement made to the performance of the MoS$_2$ device by annealing can help improve the efficiency of thin-film-based nanoelectronic devices[5, 35].

**Coexistence of 1H and 1T' phases**. To verify the phase transition process of monolayer MoS$_2$/Au, synchrotron-based PES, PL and Raman spectroscopy are measured after annealing. Figure 3a, b shows the evolution of Mo-3$d$ and S-2$p$ core-level spectra, respectively, of MoS$_2$/Au after annealing. The Mo-3$d$ spectrum of the as-prepared sample shows a 1H-MoS$_2$ doublet at ~229.7 (3$d_{5/2}$) and ~232.9 eV (3$d_{3/2}$)[14], and the S-2$s$ component at ~226.5eV[36]. New components at lower binding energies relative

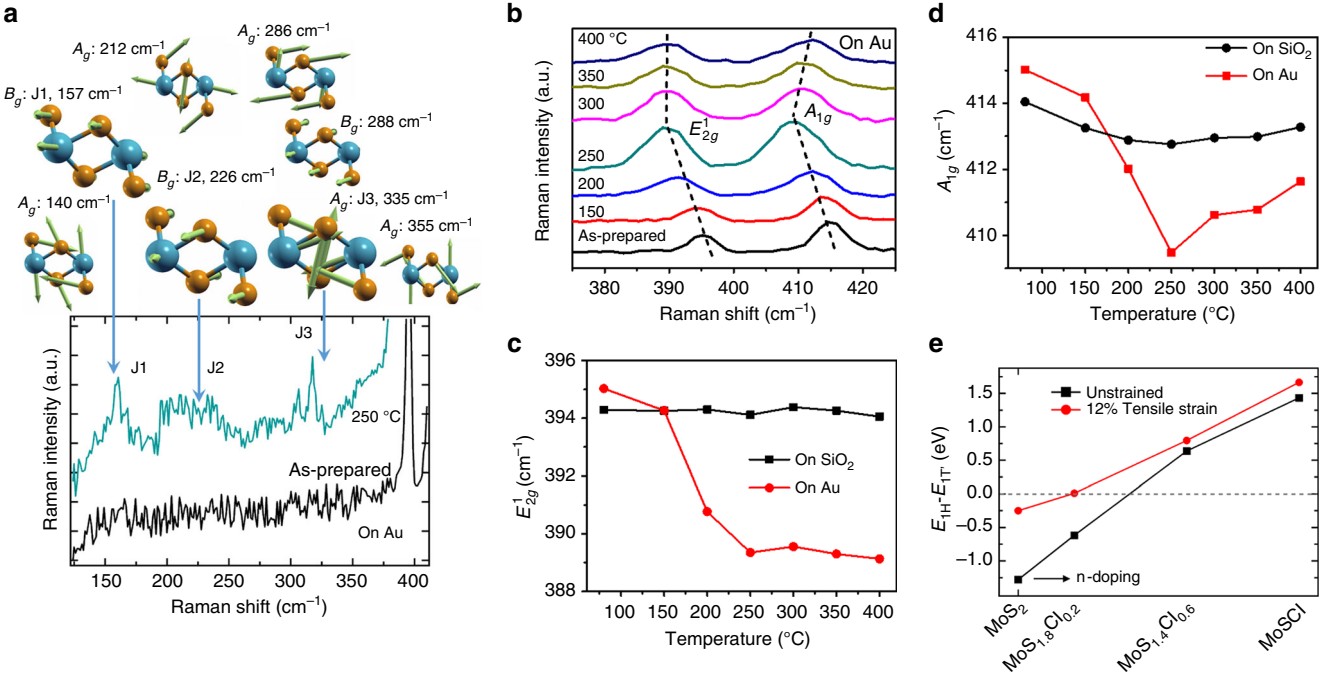

**Fig. 4** Annealing temperature dependence of Raman spectroscopy. **a** (*top*) Calculated Raman active eigenmodes and frequencies, with corresponding irreducible representation assigned from group theory. **a** (*bottom*) Comparison of Raman spectra of MoS$_2$/Au before and after annealing at 250 °C. **b** Raman spectra of MoS$_2$ film on Au electrode in the device (Fig. 2b) at various temperatures. **c, d** Extracted frequencies of $E^1_{2g}$ and $A_{1g}$ modes as functions of temperature. The errors of the extracted energy positions are 0.5 cm$^{-1}$. **e** Calculated energy difference between 1H and 1T' phases as a function of electron doping concentration for unstrained and strained (12% tensile) MoS$_2$

to the 1H-MoS$_2$ doublet grow in intensity upon annealing and are maximized at 250 °C. This indicates the formation of a new species. This suggests the thermal decomposition of MoS$_2$ $\varepsilon_1$ and $\varepsilon_2$ at 350 °C in Fig. 1e, f are different compared to other temperatures, which is also attributed to the partial decomposition of MoS$_2$.

Fitting of these peaks at various annealing temperatures in the Mo-3$d$ spectra reveals an additional doublet at ~0.7 eV below the 1H-MoS$_2$ doublet. The additional S-2$p$ doublet are found to be ~0.5 eV below the 1H-MoS$_2$ doublet at ~162.5 (S-2$p_{1/2}$) and ~163.7 eV (S-2$p_{3/2}$), respectively[36]. The parallel shift of these new doublets and the decrease in MoS$_2$/Au contact resistance suggest the formation of a new quasimetallic MoS$_2$ structure[14, 37]. Combining this with our Raman and ellipsometric studies, we assign the new doublets to the 1T' MoS$_2$ phase. Figure 3c shows the proportion of 1H-MoS$_2$ and 1T'-MoS$_2$ components of Mo-3$d_{5/2}$ peaks as functions of annealing temperature, suggesting the coexistence of 1H and 1T' phases in monolayer MoS$_2$/Au. The percentage of 1T'-MoS$_2$ is ~7% for the as-prepared sample and reaches a maximum at ~37% after annealing to 200–250 °C, consistent with the transport result (Fig. 2d). The low 1T'-phase yield explains the low mobility in the electrical measurements before annealing, and is maximized after annealing to 200–250 °C (Fig. 2d). Note that before annealing, the mid-infrared peak (Fig. 1f) is not solely the effect of the 1H–1T' phase transition. It is also an effect of strong electronic correlations.

The PL spectra of the monolayer MoS$_2$/Au after annealing are displayed in Fig. 3d along with the extracted energy positions of excitonic peak A in Fig. 3e (Supplementary Fig. 2 and Supplementary Note 2). For the as-prepared MoS$_2$/Au (Fig. 3d), excitonic peaks A and B, attributed to the direct bandgap PL from the K-point, are located at 1.89 and 2.03 eV, respectively, in good agreement with a previous report on monolayer MoS$_2$[4]. Peak A is red-shifted and broadened with rising annealing temperature.

This is opposite to the effect in a previous PL study[14] where the MoS$_2$ film transits from metallic to semiconducting phase. This supports our phase transition hypothesis.

**Underlying mechanism of 1H–1T' phase transition.** Figure 4a displays the Raman spectra of MoS$_2$/Au as prepared and upon annealing at 250 °C. For the latter spectrum, three main features at ~157 (J1), ~224 (J2) and ~320 cm$^{-1}$ (J3) are observed—these are absent from the as-prepared sample. These peaks are characteristic of the 1T' octahedral structure[22, 23], consistent with published theoretical and experimental studies that show this is the 1T'-phase rather than 1T-phase MoS$_2$[23, 38, 39]. The top of Fig. 4a displays the calculated Raman active modes and frequencies with their corresponding irreducible representation assigned from group theory of 1T'-MoS$_2$. Most of the experimental peaks correspond to the calculated Raman active eigenmodes. This is clear evidence of the presence of 1T'-MoS$_2$. The slight difference in peak positions is likely due to the tensile strain between MoS$_2$ and Au.

Figure 4b shows the Raman spectra for monolayer MoS$_2$/Au evolve after different annealing temperatures. The spectra show two main modes—the in-plane $E^1_{2g}$ mode (opposing vibrations of the in-plane Mo and S atoms) and the out-of-plane $A_{1g}$ mode (opposing vibrations of the two out-of-plane S atoms). The $E^1_{2g}$ and $A_{1g}$ modes of 1H-MoS$_2$ are still observable after annealing due to the partial phase transition process[22]. Interestingly, the frequencies of both $E^1_{2g}$ and $A_{1g}$ for MoS$_2$/Au shift to lower frequency upon annealing to 250 °C, while that for MoS$_2$ on SiO$_2$/Si remain constant (Supplementary Fig. 3). Figure 4c, d shows the extracted frequencies of the $E^1_{2g}$ and $A_{1g}$ modes as functions of annealing temperature for monolayer MoS$_2$ on Au and on SiO$_2$/Si.

As suggested in a previous study, Raman spectroscopy is an effective technique to characterize the effects of strain and charge

doping in 2D materials by correlation analysis of the Raman modes[40]. It has been demonstrated in monolayer $MoS_2$ that upon electron doping, there is a red-shift in the $A_{1g}$ mode and an increase of its full-width-half-maximum (FWHM), while the $E_{2g}^1$ mode remains unchanged[41]. This indicates that $A_{1g}$ mode is strong sensitivity to electron doping due to stronger electron–phonon coupling of the $A_{1g}$ mode[41]. Conversely, with increasing tensile (compressive) strain effect, the $E_{2g}^1$ mode is red-shifted (blue-shifted) while the $A_{1g}$ mode remains constant[42, 43]. In our case both modes are red-shifted—an indication of the concurrent increase in both tensile strain and electron doping. As shown in Fig. 4b, c, the red-shift of the $E_{2g}^1$ mode for $MoS_2$/Au from ~395 (as-prepared) to ~389 $cm^{-1}$ (250 °C), indicates an increase of tensile strain by ~2.7% after annealing at 250 °C[42]. Consistent with a previous study[44], this increase in tensile strain may be a contributing factor to the red shift and intensity quenching of peak A observed in the PL data (Fig. 3d). Meanwhile, frequency of the $A_{1g}$ mode red-shifts from ~415 to ~409 $cm^{-1}$ (Fig. 4b, d), indicating an increase in electron doping concentration by more than $1.5 \times 10^{13}$ $cm^{-2}$ from Au[41]. Our observation is consistent with a previous DFT study which has shown increasing electron affinity of $MoS_2$ with tensile strain[45]; this allows greater charge transfer from Au to $MoS_2$. It is noted that the frequency of both $A_{1g}$ and $E_{2g}^1$ modes can also be influenced by the presence of structural defects in monolayer $MoS_2$[46]. The frequency of $A_{1g}$ mode increases alongside a decrease in the frequency of $E_{2g}^1$ mode with increasing defects. However, this does not take place in our experimental observation as annealing temperature increases. Besides, the FWHM of the $A_{1g}$ mode (~ 5.7 $cm^{-1}$)and $E_{2g}^1$ mode (~ 4.6 $cm^{-1}$) of our as-prepared monolayer- $MoS_2$ are close to those of pristine $MoS_2$ sample as previously reported[46]. These allow us to rule out the defect effect as the main mechanism involved.

Doping-dependent energetics for large-strained (12%) and unstrained $MoS_2$ (Fig. 4e) also show that in the presence of tensile strain, 1T'-$MoS_2$ is energetically favored at smaller electron doping concentrations compared to the unstrained case. This shows that tensile strain and charge transfer work together to enable the phase transition proposed here. The effect of tensile strain facilitating a semiconductor-to-metal phase transition demonstrated recently in thin-film $MoTe_2$[47] further supports our conclusion. As suggested in previous theoretical studies[28, 48], having monolayer $MoS_2$ on Au alone is not a sufficient condition for the 1H–1T' phase transition. This is because of the inefficiency in electron injection due to the formation of an interfacial tunnel barrier[28, 48]. However, further studies are needed to understand why strain and electron transfer take place for $MoS_2$/Au upon annealing. One possibility is that the increase in annealing temperature reduces the tunnel barrier width and energy at the $MoS_2$/Au interface[28, 48]. This improves the electronic transparency of the contact.

Our ellipsometry results show that the inverted gap in $MoS_2$/Au narrows upon annealing to 250 °C (Fig. 1f), while the interfacial tensile strain and electron doping in $MoS_2$ increase (Fig. 4c, d). This is consistent with the previous computation study which reports the tuning of $2\delta$ of 1T'-$MoS_2$ by increasing strain[20].

Furthermore, we have considered the possibility that the metallic behavior may be due to the substitution of Au adatoms on sulfur vacancies in $MoS_2$/Au upon annealing. Our density of state calculations (Supplementary Fig. 4 and Supplementary Note 3) suggest that this effect may contribute partially to the metallic phase of the system. However, our experimental studies show that this effect is minimal due to the lack of strong binding between $MoS_2$ and Au as suggested by the Au-4f PES data in Supplementary Fig. 1d (Supplementary Note 4).

## Discussion

The collective trends observed in the absorption (changes to inverted gap, Fig. 1f), transport (Fig. 2d), PES (Fig. 3b, c), PL (Fig. 3d) and Raman spectroscopy (Fig. 4c, d) with respect to annealing temperature are strikingly consistent and show that the temperature window of 200–250 °C is the optimal condition for 1H–1T' phase transition. Results from this comprehensive study indicates that electron doping from gold, facilitated by interfacial tensile strain, leads to the partial and localized 1H–1T' phase transition in monolayer $MoS_2$ which opens a tunable inverted gap. Going forward, the quasimetallic 1T' phase and its inverted gap are potentially attractive features of 2D-$MoS_2$ for device fabrication.

## Methods

**Sample preparation.** For spectroscopic ellipsometry and PES measurements, high-quality and large-area $MoS_2$ monolayers are needed. The $MoS_2$ monolayer was synthesized on a sapphire surface by the CVD method using $MoO_3$ and S powders as the reactants[9]. A 200 nm gold film was sputtering coated on $SiO_2$/Si substrate. The CVD-grown $MoS_2$ monolayer was transferred to the Au thin film on $SiO_2$/Si substrate using polymethyl methacrylate and annealed at 80 °C to enhance the contact between the film and substrate as well as to eliminate residues. The monolayer property of $MoS_2$ grown by CVD was confirmed by spectroscopic ellipsometry measurement as shown in Fig. 1e, f. As reported in previous studies[34, 49], the absorption spectra of multilayer and bulk $MoS_2$ show a non-zero broad pre-peak associated with the indirect transition (tail at ~1.7 eV) below peaks A and B; peak C is red-shifted and broadens increasing $MoS_2$ thickness. However, these two absorption features are absent in the absorption coefficient spectrum of our $MoS_2$ on $Al_2O_3$ grown by CVD, as shown in the inset of Fig. 1f. Our data are in good agreement with the spectrum for monolayer $MoS_2$ reported in previous studies[49, 50]. This suggests that our CVD-grown $MoS_2$ in this report is monolayer. To measure electrical transport, PL and Raman spectroscopy, a $MoS_2$ field-effect transistor was required. The 5/45 nm Ti/Au electrodes on $SiO_2$/Si substrate were fabricated using standard electron beam lithography processes. The mechanical exfoliated $MoS_2$ monolayer was dry transferred onto the electrodes (Fig. 2a, b) to make a $MoS_2$/Au interface. The two Ti/Au electrodes and Si substrate formed the source, drain and gate electrodes, respectively. The monolayer property of mechanical exfoliated $MoS_2$ was confirmed by PL spectroscopy measurement. For as-prepared $MoS_2$/Au (main text Fig. 3d), excitonic peaks A and B are located at 1.89 and 2.03 eV respectively—in good agreement with report on mechanically exfoliated monolayer samples[4]. The samples and devices were annealed for 15 min at different temperatures in vacuum, and all the measurements were done after the sample cooled to room temperature.

**Spectroscopic ellipsometry.** We use a J. A. Woollam Co., Inc. VASE and IR-VASE spectroscopic ellipsometer with a photon energy of 0.4–4 and 0.07–0.17 eV, respectively, to measure the ellipsometry parameters Ψ (the ratio between the amplitude of p- and s-polarized reflected light) and Δ (the phase difference between of p- and s-polarized reflected light). The annealing process is performed in a high vacuum chamber with a base pressure of $1 \times 10^{-9}$ mbar. The absorption coefficient of $MoS_2$ monolayer was extracted from the parameters Ψ and Δ utilizing an air/$MoS_2$/Au (or $Al_2O_3$) multilayer model (see Supplementary Information), where the monolayer $MoS_2$ composed of an average homogeneous and uniform medium.

**Synchrotron-based PES.** The PES data were taken in a ultrahigh vacuum chamber with a base pressure of $1 \times 10^{-10}$ mbar at the SINS beamline of Singapore Synchrotron Light Source (SSLS). The measurements were performed immediately after the samples were cooled down to room temperature. The photon energy of 365 and 60 eV were used to probe the Mo-3d, S-2p, Au-4f and valence band spectra. The work function was measured using 60 eV photon energy with a −7 V applied bias (see Supplementary Information and Supplementary Fig. 1a). All spectra were collected at normal emission using a VG Scienta R4000 analyzer and normalized by photon current. The photon energy was calibrated using the Au-4$f_{7/2}$ core level peak at 84.0 eV of a sputter-cleaned gold foil in electrical contact with the sample. The binding energy is referred to the Fermi level of gold foil. The least-squares peak fit analysis were performed using Voigt photoemission profiles with constant Lorentzian (15%) and Gaussian (85%) line shape. For S-2p and Mo-3d spectra fitting, splitting difference of ~1.18 eV with branching ratio of 2 ($2p_{3/2}$):1 ($2p_{1/2}$) and ~ 3.15 eV with branching ratio of 3 ($3p_{5/2}$):2 ($3p_{3/2}$) were used, respectively.

**First-principles calculation.** The Raman frequencies and intensities of zone center (Γ-point) phonons are calculated using local density approximation, within density functional perturbation theory[51], as implemented in the Quantum Espresso Package[52]. The structure is considered as relaxed when the maximum component

of the Hellmann–Feynman force acting on each ion is less than 0.003 eV Å$^{-1}$. To obtain converged results, we use an energy cutoff of 65 Ry, and a Monkhorst–Pack k-point mesh of $9 \times 16 \times 1$. The monolayer is simulated by adding a vacuum thickness of 16 Å to prevent interactions between periodic image slabs. For the electron doping calculation, the $Mo(S_{1-x}Cl_x)_2$ is simulated using virtual crystal approximation as implemented in Quantum Espresso. The Cl atom has one more electron than the S, and can act as donor.

**Data availability**. The data files that support the findings of this study are available from the corresponding author on reasonable request.

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

## Acknowledgements

This work is financially supported by the National Natural Science Foundation of China (51472164), the Natural Science Foundation of SZU (000050), the 1000 Talents Program for Young Scientists of China, Shenzhen Peacock Plan (KQTD2016053112042971), the Educational Commission of Guangdong Province (2015KGJHZ006), the Science and Technology Planning Project of Guangdong Province (2016B050501005), the China Postdoctoral Science Foundation funded project (2016M600664), Singapore National Research Foundation under its Competitive Research Funding (NRF-CRP 8-2011-06 and NRF-CRP15-2015-01), MOE-AcRF Tier-2 (MOE2015-T2-1-099, MOE2015-T2-1-099

and MOE2015-T2-2-147), 2015 PHC Merlion Project and FRC (R-144-000-368-112, R-144-000-346-112 and R-144-000-364-112). We thank Centre for Advanced 2D Materials and Graphene Research Centre at National University of Singapore to provide the computing resource. We acknowledge the Singapore Synchrotron Light Source (SSLS) for providing the facility necessary for conducting the research. The SSLS is a National Research Infrastructure under the National Research Foundation Singapore.

## Author contributions

X.Y., L.M.W., S.J.W. and C.S.T. performed spectroscopic ellipsometry measurements; Q.W. and W.Z. prepared high-quality monolayer films, and performed transport, Raman and photoluminescence spectroscopy measurements; X.Y. and L.C. performed synchrotron-based photoemission spectroscopy measurement; X.L. and S.Y.Q. carried out the first-principles calculation; Y.Z. and S.Y.Q. performed metallic projected density of states calculation; X.Y., W.Z., A.R. and A.T.S.W. analyzed the data and wrote the manuscript with the assistance from C.S.T., X.L., L.C., Y.Z., S.Y.Q., L.M.W., S.J.W. and Q.W.; W.Z., A.R. and A.T.S.W. supervised the project.

## Additional information

**Competing interests:** The authors declare no competing financial interests.

