## [Peer Review File · Nature Communications]

Reviewers' Comments:

Reviewer #1:

Remarks to the Author:

This work reports the observation of inverted gap in distorted MoS₂ monolayers. The presence of inverted gap was suggested by absorption and ellipsometry measurements. 1T' structure of 1L-MoS₂ was assessed by XPS, Raman and FET device. I find that the phase transition from 2H to 1T' of 1L-MoS₂ by interfacing with Au and annealing is convincingly shown here. However the unique finding and the significance of this work should lie in experimental observation and tunability of the inverted gap because the structural transition of MoS₂ by doping electrons is already well known. In this regard, I think authors should show the presence (and maybe tunability as well) of fundamental gap to be considered for publication in Nat. Commun. Presence of fundamental gap (~0.08 eV) would be much more specific and unambiguous while optical transition below 1 eV can appear with many other reasons in monolayer TMDs and the claimed correlation between rising of peak below 1eV and above 2.4 eV is not so clear.

Reviewer #2:

The authors address quite interesting phenomena that was observed from MoS₂ on Au electron at moderate temperature. The synthesis of 1T'-MoS₂ has been demonstrated mostly through the chemical treatment. But, according to the manuscript, 1T'-MoS₂ can be easily formed on the surface of metal with a heat treatment. However, the reviewer found several issues need to be clarified to meet the scientific standard for Nature Communication, which are listed below.

1. A similar work has been reported [Y. Kang et al, Adv. Mater. 26, 6467 (2014)], but the claims made by two different groups of authors are different. The author in the manuscript report that the formation of 1T' MoS₂ is observed, although the previous paper reported the formation of 1T phase. In the course of the confirmation of the structure, Raman spectroscopy was used in both papers. The similar peaks (J1, J2, and J3) from the spectra are used in the manuscript as the evidence for the existence of 1T' phase. It is very necessary for the authors to prove what kind of MoS₂ phase of is formed through the annealing process because the formation of 1T' MoS₂ is the initiation of the discussion.
2. In the synthesis of 1T' using Li intercalations, the transition from 1T to 1T' phase is involved with layer-to-layer interaction or the adsorption of organic functional groups [J. Am. Chem. Soc, 121, 638 (1999), Chem. Mater. 9, 879 (1997), J. Phys. Chem. C 118, 1515 (2014), Curr. Appl. Phys. 17, 60 (2017)]. In the formation of 1T' phase of MoS₂ in the manuscript, how is it different from previous chemical treatments.
3. The authors claim in the manuscript "the metallic behavior may be due to the substitution of Au adatoms on Sulphur vacancies.....". It is reasonable to say that. If Au atoms

interact with Mo atoms, then, the atomic binding between Mo and S will be weakened, resulting in the formation of S vacancies [J. Kang et al, Phys. Rev. X 4, 031005 (2014)]. In addition, the density of natural S vacancies in MoS₂ is very high (an order of 10¹²/cm²) [H.

Qiu et al, Nature Comm. 4, 2642(2013)]. If this is true, there should be a peak related with S vacancies in PES in Fig. 3(a). Why is it not observable in the figure even after annealing.

4. From the interpretation of Raman spectra in Fig. 4b-d, the A_{1g} and E_{2g1} peak shifts are signified as an evidence of doping and stress on MoS₂ on Au electrode. Can the author separate these two effects? As it is known that the effects of doping and strain are coupled in graphene [Nature Comm, 3, 1024 (2012)], it is important to quantify the effect of doping and strain in separate in order to understand the phase transition of MoS₂ on Au metal.

5. In the electrical measurements, the field effect mobility of untreated MoS₂ is very low (0.0041 cm²

/Vs), although it enhances considerably after the thermal treatment. However, in

Fig. 1f, the existence of 1T' phase, probed by dielectric function ϵ_2 , is already significant before the thermal annealing. Therefore, the field effect mobility of untreated MoS₂ should not be that small. Can the authors explain the reason?

We are very grateful for the critical and constructive comments provided by both referees 1 and 2. We have addressed all the issues raised and made corresponding improvements, and we believe that this manuscript is now ready to be considered for publication in *Nature Communications*.

Our point-by-point responses to the reviewers' comments are as follows.

Note: All Referees comments/questions are in black. Our responses to them, new discussions/Figures Caption/Tables in the revised manuscript are all in blue.

Reviewers' comments:

Reviewer #1 (Remarks to the Author):

This work reports the observation of inverted gap in distorted MoS₂ monolayers. The presence of inverted gap was suggested by absorption and ellipsometry measurements. 1T' structure of 1L-MoS₂ was assessed by XPS, Raman and FET device. I find that the phase transition from 2H to 1T' of 1L-MoS₂ by interfacing with Au and annealing is convincingly shown here.

We are thankful for the referee's appreciation of our experimental findings and the convincing evidence of the 1H-1T' phase transition in monolayer-MoS₂ on Au film.

However the unique finding and the significance of this work should lie in experimental observation and tunability of the inverted gap because the structural transition of MoS₂ by doping electrons is already well known.

We agree with the reviewer that the unique finding and the significance of this work is in the experimental observation and tunability of the inverted gap. We therefore further emphasize this in our revised manuscript.

In this regard, I think authors should show the presence (and maybe tunability as well) of fundamental gap to be considered for publication in Nat. Commun. Presence of fundamental gap (~0.08 eV) would be much more specific and unambiguous ...

Following the referee's suggestion, we perform additional spectroscopic ellipsometry experiments in the far-infrared regime to detect the associated fundamental gap. The results are shown in Fig. R1. We observe three important phenomena. Firstly, the fundamental gap at ~0.10 eV is indeed observed upon sample annealing. Secondly, the fundamental gap is

annealing-temperature independent, and this is consistent with previous theoretical calculation for the case of MoS₂ [Supplementary Fig. S9a in *Science* **346**, 1344-1347 (2014)]. Thirdly, the Drude response is enhanced upon sample annealing. The enhanced Drude response is due to charge transfer and is responsible for the observed increased mobility.

Figure R1. The dielectric function ϵ_2 of MoS₂ as a function of annealing temperature with colour-coded arrows indicating the position of the fundamental gap.

Changes made in the revised manuscript:

We include the above discussion and figure (c.f. Fig. 1i) in the revised manuscript.

‘Furthermore, another interesting peak is observed in the far-infrared regime at ~ 0.10 eV upon sample annealing at 250 °C as shown in Fig. 1i. This is ascribed to the fundamental gap as discussed later.’

‘Moreover, the feature which appears upon sample annealing at ~ 0.10 eV (Fig. 1i) is associated with the fundamental gap of 1T’-MoS₂ (c.f., Fig. 1d). It is nearly annealing-temperature independent after it appears. This is consistent with the previous theoretical study of monolayer-MoS₂. Experimental data in Fig. 1i also shows that the Drude response enhances upon sample annealing. It is responsible for the increased mobility due to charge-transfer, which will be further demonstrated.’

Additional caption: **i**, Imaginary component of dielectric function, ϵ_2 , of monolayer-MoS₂/Au in the far-infrared regime from IR-VASE spectroscopic ellipsometry at various annealing temperatures with colour-coded arrows indicating the position of the fundamental gap.

... while optical transition below 1 eV can appear with many other reasons in monolayer TMDs ...

The referee raises interesting points with regard to the inverted and fundamental gaps.

Now, with the new set of data (Fig. R1), we provide direct evidence for the coexistence of both the inverted gap 2δ and fundamental gap E_g in the MoS₂/Au sample after annealing.

We highlight that an inverted gap ($\sim 0.5\text{eV}$) is theoretically predicted in *Science* **346**, 1344-1347 (2014) for the MoS₂ in 1T'-phase, and is experimentally observed in this work. Beside the observation of a mid-infrared peak at $\sim 0.5\text{eV}$, this peak position can also be tuned as a function of strain (cf. Figs. 1f and 4b,c). This is also consistent with the theoretical calculation of the value of the inverted gap with varying strain presented in the supplementary of *Science* **346**, 1344-1347 (2014). Therefore, we can convincingly conclude that the mid-infrared transition peak at $\sim 0.5\text{eV}$ in our data (c.f. Fig. 1f) indeed corresponds to the inverted gap of 1T'-MoS₂. The presence of this inverted gap is further substantiated by the PES measurement of the valence band spectra (Fig. S1c in supplementary).

Furthermore, to address the referee's concern, we also rule out other possibilities for optical transitions below 1 eV.

1. **Vacancies.** The presence of intrinsic vacancies in CVD grown monolayer-MoS₂ has been reported [Nano Lett., 2013, 13 (6), pp 2615–2622; Nanotechnology. 2014 Sep 19;25(37):375703] Should vacancies be the contributing factor leading to the optical response below 1eV, there should be a peak in the MoS₂/Al₂O₃ data as well (main text Fig. 1f). However, the optical transition appears only after the transfer of MoS₂ onto Au [Applied Physics Express 6, 125801 (2013)]. Therefore, we can rule out the contribution of vacancies to the optical transition at $\sim 0.5\text{eV}$.
2. **Impurities.** With increasing annealing temperature in ultra-high vacuum, the physisorbed impurities on the MoS₂ sample should be reduced, thereby causing a reduction to any impurity-induced optical transition. However, instead of a reduction, we see an evident rise in the optical transition at $\sim 0.5\text{eV}$ with increasing annealing temperature. This shows that physisorbed impurities should not be the contributing factor leading to this sub-1eV optical response.

Change made in the revised manuscript:

We include the above discussion in the revised manuscript.

'We examine other possible explanations for the mid-infrared optical peak at $\sim 0.5\text{eV}$ in Fig. 1f. Namely, vacancies introduced during sample growth, and physisorbed impurities introduced during sample transfer. Firstly, the presence of intrinsic vacancies in CVD-grown monolayer-MoS₂ has been reported. However, this can be ruled out as the cause of the $\sim 0.5\text{eV}$ optical peak because this peak only appears after the transfer of MoS₂ onto Au and is absent in the case of MoS₂/Al₂O₃. Secondly, there is an increase in the $\sim 0.5\text{eV}$ optical peak intensity with increasing annealing temperature. However, sample annealing in ultra-high vacuum results in the desorption of absorbed impurities. Hence, we can also rule out physisorbed impurities being introduced during the transfer-process as a contributing factor to the $\sim 0.5\text{eV}$ mid-infrared peak.'

... and the claimed correlation between rising of peak below 1eV and above 2.4 eV is not so

clear.

We would like to clarify that in Fig. 1f, there is a significant decrease in spectral-weight at the high-energy region (above 2.5eV) when MoS₂ is transferred from Al₂O₃ (dashed violet line) to Au (black solid line) as shown in Figure 1f. It gives us important information on the so-called strong electronic correlations. Even before the high-temperature annealing process, this significant drop in spectral-weight cannot be explained solely by the 1T'-lattice change because of the low yield in 1H-1T'-MoS₂ transition (~7% based on PES measurement as discussed in Figs. 3a & b before annealing).

According to the Zaanen-Sawatzky-Allen model [*Phys. Rev. Lett.* **55**, 418-421 (1985)], spectral-weight transfers can occur over a broad energy range due to strong local charge interactions in correlated electronic systems. Therefore, the large spectral-weight transfer from the high-energy region (above 2.5eV) to the inverted gap energy region (below 1.5eV) reveals that electron-electron correlation must play an important role in the origin of this inverted gap. This observation is also supported by a previous GW-based study of the 1T'-phase 2D-TMDs [*Science* **346**, 1344-1347 (2014)]. Therefore, by combining the predictions of this theoretical study with our optical data, we deduce that charge-lattice coupling is the primary cause of this inverted gap.

Change made in the revised manuscript:

We include the above discussion in the revised manuscript.

'An important observation from our spectroscopic ellipsometry data (Fig. 1f) is the significant decrease in spectral-weight at the high-energy region (above 2.5eV) when MoS₂ is transferred from Al₂O₃ (dashed violet line) to Au (black solid line). Even before the high-temperature annealing process, this significant drop in spectral-weight cannot be explained solely by the change in 1T' lattice-structure because of the low yield in 1H-1T' MoS₂ transition (~7% as discussed later). According to the Zaanen-Sawatzky-Allen Theory, spectral-weight transfers can occur over a broad energy range due to strong local charge interactions in correlated electronic systems.'

Reviewer #2

The authors address quite interesting phenomena that was observed from MoS₂ on Au electron at moderate temperature. The synthesis of 1T'-MoS₂ has been demonstrated mostly through the chemical treatment. But, according to the manuscript, 1T'-MoS₂ can be easily formed on the surface of metal with a heat treatment. However, the reviewer found several issues need to be clarified to meet the scientific standard for Nature Communication, which are listed below.

1. A similar work has been reported [Y. Kang et al, Adv. Mater. 26, 6467 (2014)], but the claims made by two different groups of authors are different. The author in the manuscript report that the formation of 1T' MoS₂ is observed, although the previous paper reported the formation of 1T phase. In the course of the confirmation of the structure, Raman spectroscopy was used in both papers. The similar peaks (J1, J2, and J3) from the spectra are used in the manuscript as the evidence for the existence of 1T' phase. It is very necessary for the authors to prove what kind of MoS₂ phase of is formed through the annealing process because the formation of 1T' MoS₂ is the initiation of the discussion.

We thank the referee for the constructive suggestions. In the work by Y. Kang et al (Adv. Mater. 26, 6467 (2014)), Au nanoparticles were deposited on monolayer-MoS₂. They then indicated that this process might induce a transient reversible 1H-to-1T phase transition in MoS₂, which was monitored by Raman and photoluminescence spectroscopies.

However, we wish to clarify that recent reports have shown both theoretically and experimentally that the 1T-structure is unstable in free-standing conditions and will spontaneously relax to a distorted 1T' structure [*Science* 346, 1344-1347 (2014); *Nat Phys* 11, 482-486 (2015); *J. Am. Chem. Soc.*, **2017**, 139 (6), pp 2504–2511]. Furthermore, it has been demonstrated both theoretically and experimentally that only the 1T'-phase shows J1,J2,J3 peaks in the Raman spectrum, [*Nano Lett.*, 2015, 15 (1), pp 346–353; *J. Am. Chem. Soc.* 2017, 139, 2504–2511; *J. Chem. Phys.* 2013, 139 (17), 174702] but not the 1T-phase [*J. Chem. Phys.* 2013, 139 (17), 174702].

Change made in the revised manuscript:

The following has been added to the main text to better highlight that J1, J2 and J3 are the result of 1T'-phase MoS₂ and not 1T-phase. References have also been added to support our statement.

'These peaks are characteristic of the 1T' octahedral structure, consistent with published theoretical and experimental studies that show this is the 1T' - rather than 1T-phase MoS₂.

2. In the synthesis of 1T' using Li intercalations, the transition from 1T to 1T' phase is involved with layer-to-layer interaction or the adsorption of organic functional groups [J. Am. Chem. Soc, 121, 638 (1999), Chem. Mater. 9, 879 (1997), J. Phys. Chem. C 118, 1515 (2014), Curr. Appl. Phys. 17, 60 (2017)]. In the formation of 1T' phase of MoS₂ in the manuscript, how is it different from previous chemical treatments.

We thank the referee for this query to help improve our manuscript. The physical mechanism of synthesizing 1T' using Li-intercalation is based on electron-doping [Nature Nanotechnology 10, 313–318 (2015)]. In our case, the underlying physical mechanism is not identical.

We demonstrate in our work that electron-doping from gold is facilitated by interfacial tensile strain to induce the 1H-1T' phase transition. Previous computational studies have suggested that solely having monolayer-MoS₂ on Au is not a sufficient condition for the 1H-1T' phase transition due to the inefficiency in electron injection [PRL 108, 156802 (2012); PRX 4, 031005 (2014)]. Our experimental study shows that the process of annealing results in the increase of both interfacial strain and electron-doping (Fig. 4), which together results in the phase transition. Our theoretical calculations further support the hypothesis that increasing interfacial tensile strain can better facilitate the 1H-1T' phase transition at lower electron-doping level.

In addition, the method we employ is a convenient and straightforward way of inducing a semiconductor-to-metal phase transition in 2D-MoS₂ based on an annealing process. We would like to highlight that this process may have potential applications such as in the fabrication of 2D-TMD Field-Effect Transistors.

Change made in the revised manuscript:

We address this point in separate sections of the manuscript by highlighting the similarity and the major differences involved between the chemical techniques used in other studies and the annealing-based technique described in this work.

'In this regard, controlling the phase composition of 2D-MoS₂ is important for device integration and scalable processing. The semiconductor-to-metal phase-transition of 2D-TMDs was previously demonstrated via *n*-butyl lithium (*n*-BuLi) treatment by electron-transfer from *n*-BuLi to TMD and the interlayer atomic plane gliding during intercalation [ref, Nature Nanotechnology 10, 313–318 (2015)].'

'Detailed analysis shows that the 1H-1T' phase-transition is due to electron-transfer from gold to monolayer-MoS₂. While this underlying mechanism is similar to the previously reported *n*-butyl lithium (*n*-BuLi) treatment method, the phase transition of monolayer-MoS₂ on Au is further facilitated by interfacial strain. This is a convenient and straightforward way of inducing a semiconductor-to-metal phase transition in 2D-MoS₂ based on an annealing

process. This technique has potential applications such as the fabrication of 2D-TMD-based Field-Effect Transistors.’

‘As suggested in previous theoretical studies, having monolayer-MoS₂ on Au alone is not a sufficient condition for the 1H-1T’ phase transition. This is because of the inefficiency in electron injection due to the formation of an interfacial tunnel barrier [Phys Rev Lett 108, 156802 (2012)].’

The following references have also been added to the manuscript:

[J. Am. Chem. Soc, 121, 638 (1999), Chem. Mater. 9, 879 (1997), J. Phys. Chem. C 118, 1515 (2014), Curr. Appl. Phys. 17, 60 (2017)]

3. The authors claim in the manuscript “the metallic behavior may be due to the substitution of Au adatoms on Sulphur vacancies.....”. It is reasonable to say that. If Au atoms interact with Mo atoms, then, the atomic binding between Mo and S will be weakened, resulting in the formation of S vacancies [J. Kang et al, Phys. Rev. X 4, 031005 (2014)]. In addition, the density of natural S vacancies in MoS₂ is very high (an order of 10¹²/cm²) [H. Qiu et al, Nature Comm. 4, 2642 (2013)]. If this is true, there should be a peak related with S vacancies in PES in Fig. 3(a). Why is it not observable in the figure even after annealing.

We are very grateful to the referee for highlighting the role of Sulphur vacancies. Note that PES has a surface sensitivity of about 1%. Since the natural S vacancies in MoS₂ is ~10¹² cm⁻² and the estimated sulphur atomic density in monolayer-MoS₂ is ~10¹⁵ cm⁻², the percentage S vacancies is only ~0.1% – well below the PES detection limit. There should be some features related to S vacancies, but it could be not observable in PES.

4. From the interpretation of Raman spectra in Fig. 4b-d, the A_{1g} and E_{2g}¹ peak shifts are signified as an evidence of doping and stress on MoS₂ on Au electrode. Can the author separate these two effects? As it is known that the effects of doping and strain are coupled in graphene [Nature Comm, 3, 1024 (2012)], it is important to quantify the effect of doping and strain in separate in order to understand the phase transition of MoS₂ on Au metal.

The referee raises an important point. Based on our comprehensive study here, we find a strong coupling between doping (charge) and strain (lattice). This is somewhat similar to the case of graphene as pointed out correctly by the referee. In fact, we believe that because of such a complexity this system contains rich physics.

It has been demonstrated in monolayer-MoS₂ that upon electron-doping, there is a red-shift in the A_{1g}-mode while the E_{2g}¹-mode remains unchanged [Phys Rev B 85, 161403 (2012)]. This indicates that A_{1g}-mode is strong sensitivity to electron-doping due to stronger electron-phonon coupling of the A_{1g}-mode. Conversely, with increasing tensile (compressive) strain effect, the E_{2g}¹-mode is red-shifted (blue-shifted) while the A_{1g}-mode remains constant [Phys Rev B 87, 081307 (2013); ACS Nano 7, 7126-7131 (2013)].

In our case, both the A_{1g} and E_{2g}^1 -modes are red-shifted – an indication of the increase in both tensile strain and electron-doping [Nature Comm, 3, 1024 (2012)]. As shown in main text Figs. 4b,c, the red-shift of the E_{2g}^1 -mode for MoS₂/Au from ~395 (as-prepared) to ~389cm⁻¹ (250°C) indicates an increase of tensile strain by ~2.7% after annealing at 250°C. This is consistent with the result observed in PL data (main text Fig. 3c), which shows a red-shift and intensity quenching of peak A. Meanwhile, frequency of the A_{1g} -mode red-shifts from ~415 to ~409cm⁻¹ (Figs. 4b,d), indicating an increase in electron-doping concentration by more than 1.5×10^{13} cm⁻² from Au.

As explained in our response to question 2 raised by the referee, the experimental and theoretical results in our work show that the increase of interfacial strain and electron-doping together results in the 1H-1T' phase transition. The increasing interfacial strain can better facilitate the phase transition at lower electron-doping level.

Change made in the revised manuscript:

We include the above point in our revised manuscript.

‘As suggested in a previous study, Raman spectroscopy is an effective technique to characterize the effects of strain and charge-doping in 2D-materials by correlation analysis of the Raman modes. It has been demonstrated that in monolayer-MoS₂ that upon electron-doping, there is a red-shift in the A_{1g} mode while the E_{2g}^1 mode remains unchanged. This indicates that A_{1g} mode is strong sensitivity to electron-doping due to stronger electron-phonon coupling of the A_{1g} -mode. Conversely, with increasing tensile (compressive) strain effect, the E_{2g}^1 -mode is red-shifted (blue-shifted) while the A_{1g} -mode remains constant. Conversely, the E_{2g}^1 mode is sensitive to strain in monolayer-MoS₂ while the A_{1g} mode remains constant. In our case both modes are red-shifted – an indication of the concurrent increase in both tensile strain and electron-doping.’

5. In the electrical measurements, the field effect mobility of untreated MoS₂ is very low (0.0041 cm²/Vs), although it enhances considerably after the thermal treatment. However, in Fig .1f, the existence of 1T' phase, probed by dielectric function ϵ_2 , is already significant before the thermal annealing. Therefore, the field effect mobility of untreated MoS₂ should not be that small. Can the authors explain the reason?

We are indeed very grateful to the referee for highlighting this very constructive question to help us better explain the underlying physics in our manuscript. In our PES result, we have demonstrated that the yield of 1T'-phase MoS₂ is ~7% as-prepared, and it changes to ~37% after annealing to 200-250°C. The low 1T'-phase yield (~7%) and physisorbed impurities on the surface results in the low field effect mobility of untreated MoS₂ in the electrical measurements as observed.

As explained in our revised manuscript, we attribute the significant peak at $\sim 0.5\text{eV}$ before the thermal annealing to the coupling of 1T' lattice transition and strong electron correlations.

Changes made in the revised manuscript:

We have added the following to the manuscript to better explain our reasoning:

‘An important observation from our spectroscopic ellipsometry data (Fig. 1f) is the significant decrease in spectral-weight at the high-energy region (above 2.5eV) when MoS_2 is transferred from Al_2O_3 (dashed violet line) to Au (black solid line). Even before the high-temperature annealing process, this significant drop in spectral-weight cannot be explained solely by the change in lattice of the 1T'-structure because of the low yield in 1H-1T'- MoS_2 transition ($\sim 7\%$ as discussed later). According to Zaanen-Sawatsky-Allen Theory, spectral-weight transfers can occur over a broad energy range due to strong local charge interactions in correlated electronic systems.’

‘The percentage of 1T'- MoS_2 is $\sim 7\%$ for the as-prepared sample and reaches a maximum at $\sim 37\%$ after annealing to $200\text{-}250^\circ\text{C}$, consistent with the transport result (Fig. 2d). The low 1T'-phase yield explains the low mobility in the electrical measurements before annealing and is maximized after annealing to $200\text{-}250^\circ\text{C}$ (Fig. 2d). Note that before annealing, the mid-infrared peak (Fig. 1f) is not solely the effect of 1H-1T' phase transition. It is also an effect of strong electronic correlations.’

Reviewers' Comments:

Reviewer #1:

Remarks to the Author:

I think authors fully addressed the issues I raised and made convincing improvements. I can now recommend the publication of this article.

Reviewer #2:

Remarks to the Author:

All the concerns raised by the referee are well addressed in a revised manuscript. The reviewer considers that the arguments in the revised version are sound and solid in logical. The manuscript is now publishable for the journal without further revisions.

Our point-by-point responses to the reviewers' comments are as follows.

Note: All Referees comments/questions are in black. Our responses to them, new discussions/Figures Caption/Tables in the revised manuscript are in blue.

Reviewers' comments:

Reviewer #1 (Remarks to the Author):

I think authors fully addressed the issues I raised and made convincing improvements. I can now recommend the publication of this article.

We thank the referee for the constructive suggestions to improve our manuscript and are very grateful for the support for publication of this article.

Reviewer #2 (Remarks to the Author):

All the concerns raised by the referee are well addressed in a revised manuscript. The reviewer considers that the arguments in the revised version are sound and solid in logical. The manuscript is now publishable for the journal without further revisions.

We are very grateful for the referee's appreciation of our manuscript and the support given for the publication of this manuscript.